:◌: PLOS | ONE

# Ecohydrology of urban trees under passive and active irrigation in a semiarid city

**Anthony M. Luketich**[1,2º] *, **Shirley A. Papuga**[2,3º], **Michael A. Crimmins**[4]

**1** Department of Biology, University of New Mexico, Albuquerque, New Mexico, United States of America,
**2** School of Natural Resources and the Environment, University of Arizona, Tucson, Arizona, United States of America, **3** Department of Geology, Wayne State University, Detroit, Michigan, United States of America,
**4** Department of Environmental Science, University of Arizona, Tucson, Arizona, United States of America

º These authors contributed equally to this work.
* luketich.anthony@gmail.com

**Data Availability Statement:** Data was deposited to Dryad with the full citation: Luketich, Anthony (2019), Ecohydrology of urban trees under passive and active irrigation in a semiarid city, v3, Dryad, Dataset, https://doi.org/10.5061/dryad.m0cfxpnzf.

## Abstract

The infiltration of stormwater runoff for use by urban trees is a major co-benefit of green infrastructure for desert cities with limited water resources. However, the effects of this passive irrigation versus regular, controlled moisture inputs, or active irrigation, is largely unquantified. We monitored the ecohydrology of urban mesquite trees (*Prosopis spp.*) under these contrasting irrigation regimes in semiarid Tucson, AZ. Measurements included soil moisture, sap velocity, canopy greenness, and leaf-area index. We expected both irrigation types to provide additional deep (>20 cm) soil moisture compared to natural conditions, and that trees would depend on this deep moisture for transpiration and phenological activity. Results show that active irrigation supported higher soil moisture throughout the study than passive irrigation. Passive irrigation only provided additional deep moisture when green infrastructure features received impervious runoff from a city street. Sap velocity and greenness were similar under both irrigation types, outside of isolated periods of time. These differences occurred during the extremely wet summer 2017 when passively irrigated trees exhibited a greenness peak, and the dry conditions of spring when actively irrigated trees had higher sap flow and relative greenness. Finally, it was not determined that deep soil moisture had a stronger relationship with mesquite productivity than shallow moisture, but both relationships were stronger in the spring, before summer rains. This study aims to contribute empirical observations of green infrastructure performance for urban watershed management.

## Introduction

Green infrastructure such as rain gardens and bioswales are effective stormwater control measures that mitigate urban runoff and the resulting degradation of regional watersheds [1]. In addition to this main goal, green infrastructure and its concurrent vegetation provides many co-benefits to city residents, such as improved thermal comfort, air quality, and aesthetics [2,3]. In desert cities however, maintaining vegetated landscapes presents a tradeoff between

**Funding:** SP received funding from NSF CNH award 1518376. https://www.nsf.gov/awardsearch/showAward?AWD_ID=1518376. The funders had no role in study design, data collection and analysis, decision to publish, or preparation of the manuscript.

**Competing interests:** The authors have declared that no competing interests exist.

ecosystem services and the use of scarce water resources for irrigation [4,5]. This has prompted many stormwater professionals to adopt passive water harvesting as a method of urban forest irrigation [6,7]. The infiltration and retention of stormwater into the pervious green infrastructure soils allows for subsequent evapotranspiration and hence a water source for plants [8]. In some cases, passive irrigation can replace the need for regular, controlled moisture inputs—i.e. active irrigation—thus reducing demand on potable or reclaimed water supplies [9]. This is good news for water-limited cities, since urban trees can require a significant amount of irrigation for transpiration and related growth and productivity [10,11]. While research of green infrastructure performance provides mixed results [12,13], passive irrigation has shown to support increased growth rates for trees [14,15]. Still, research is needed to understand how passive green infrastructure can best support the urban forest [16].

Urban forest productivity is governed by a combination of human and environmental factors [17,18], yet soil moisture as one of these limiting factors is less well defined. Water stress is more readily considered in meteorological terms such as vapor pressure deficit, but this could be driven in large part by dry soil conditions—a recognized impediment to tree health in urban environments [12,19]. Active irrigation is common to confront this problem, especially in desert regions where the practice can decouple urban forest productivity from regional or adjacent ecosystems that are limited by variable precipitation events [20,21]. This applies to passive irrigation too, whereby impervious catchments can direct large volumes of stormwater to vegetation and in turn increase plant productivity [22]. Nevertheless, observations on the differential impacts of passive versus active irrigation on urban forest productivity have been largely unquantified.

More so than urban research, dryland ecohydrology details the explicit controls that soil moisture exhibits over plant productivity in water-limited ecosystems [23]. Unirrigated vegetation in these regions rely on infrequent and discrete pulses of precipitation [24,25]. These rain events need to be large or frequent enough to deliver moisture to depths in the soil profile that are beyond the reach of rapid atmospheric evaporation, which generally occurs >20 cm below the surface [26]. This deep moisture is critical in supporting carbon uptake and transpiration in these deserts [27–29]. Conversely, small rainstorms (<8 mm) make up the majority of annual events but rarely infiltrate to soil depths that facilitate plant uptake before evaporation occurs [30,31].

In addition to depth, soil moisture temporal dynamics play key roles in desert functioning at the plant and ecosystem scale [32–34]. For example, the water-limited southwest United States experiences peak primary productivity during summer rains, but an extended period of abundant soil moisture in the spring—usually following a wet winter—can increase carbon uptake earlier in the year [35–37]. Continuity of available moisture may also elicit distinctive plant functional responses, where for example mesquite trees (*Prosopis spp.*) with reliable access to groundwater exhibited higher annual photosynthetic activity, irrespective of rain events, compared to adjacent upland mesquites that relied on intermittent soil wetting from rain [38].

Grounded in large part on the tenets and observations of dryland ecohydrology, here we attempt to draw similarities between the natural and urban systems described above. The goal of this study is to better understand the impacts of contrasting irrigation styles—passive versus active—on soil moisture regimes and in turn urban forest productivity. We monitored the ecohydrology of green infrastructure systems in a semiarid city including canopy and water-use metrics of ornamental mesquite trees. To help predict how irrigation might influence our trees, we apply a hydrologically defined two-layer soil moisture framework [26,29]. The framework (Fig 1A) illustrates how desert vegetation in unirrigated systems often relies on deep (>20 cm) soil moisture from large (>8 mm) rains for increased productivity. We then add to

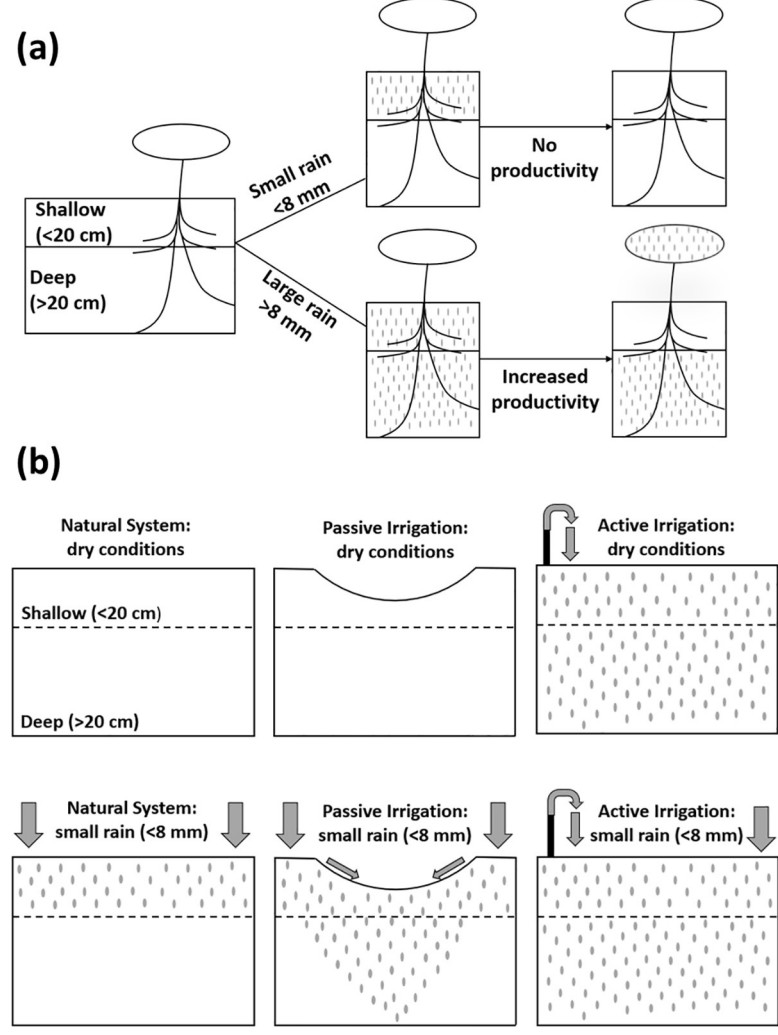

**Fig 1. Conceptual framework.** (a) Two-layer soil moisture framework for a natural dryland system: plant productivity is enhanced by deep (>20 cm) soil moisture following large (>8 mm) rains. (b) Soil moisture conditions under different irrigation styles: passive irrigation provides deep soil moisture during small rain events (<8 mm) whereas active irrigation provides perennial deep soil moisture.

this framework (Fig 1B) for a city where both passive and active irrigation are used to artificially increase cases of deep soil moisture relative to natural systems and relative to each other. Fig 1B shows the ability of green infrastructure such as a bioswale to enhance stormwater infiltration by increasing catchment and thus providing deep moisture during small rain events (e.g. passive irrigation). Conversely, active irrigation provides perennial deep soil moisture irrespective of rain events. Using these ecohydrological frameworks we address three hypotheses: 1) both passive and active irrigation will increase deep (>20 cm) soil moisture relative to a natural system; 2) mesquite trees in semiarid urban settings will depend on deep moisture that is beyond the reach of rapid atmospheric evaporation (>20 cm) for increased productivity; and 3) irrigation style can alter the temporal dynamics of mesquite ecohydrology. This study aims to contribute empirical observations of green infrastructure performance for urban watershed management and sustainable desert cities.

## Methods

### Study sites

This research took place in the semiarid city of Tucson, Arizona located within the Sonoran Desert of the southwest United States (Fig 2). The population in 2010 was 520,000 with 980,000 residents in the greater metropolitan region [39]. Tucson has a mean annual temperature of 22°C (71°F) though summer highs often exceed 38°C (100°F); mean annual rainfall is 283 mm (11.15 in) and generally occurs in a bimodal precipitation pattern [40]. The summer rainy season delivers the majority of annual precipitation to the region as high intensity, localized storms; a more variable winter rainy season occurs as lower intensity, widespread precipitation [41].

The trees investigated here are of the genus *Prosopis*—ubiquitous in the landscaping of Southern Arizona because of their tolerance for arid conditions [42]. These ornamental mesquite species readily hybridize (e.g. *P. chilensis x P. velutina*) in urban landscapes [43], making it necessary in the present study to generalize trees as *Prosopis spp*.

Data were collected at two sites where either passive irrigation (PI) or active irrigation (AI) was the primary water source for mesquite trees at each respective site. The PI site is located at the facilities of the local non-governmental organization Watershed Management Group (Fig 2D). The property (32°14'N, 110°54'W) is comparable to two single-family residences and incorporates passive green infrastructure (e.g. basins, bioswales, etc.) to encourage infiltration of stormwater. These features are the primary irrigation source for eight mesquite trees, and hence comprise the PI trees of the present study. Additional distinction among PI trees were

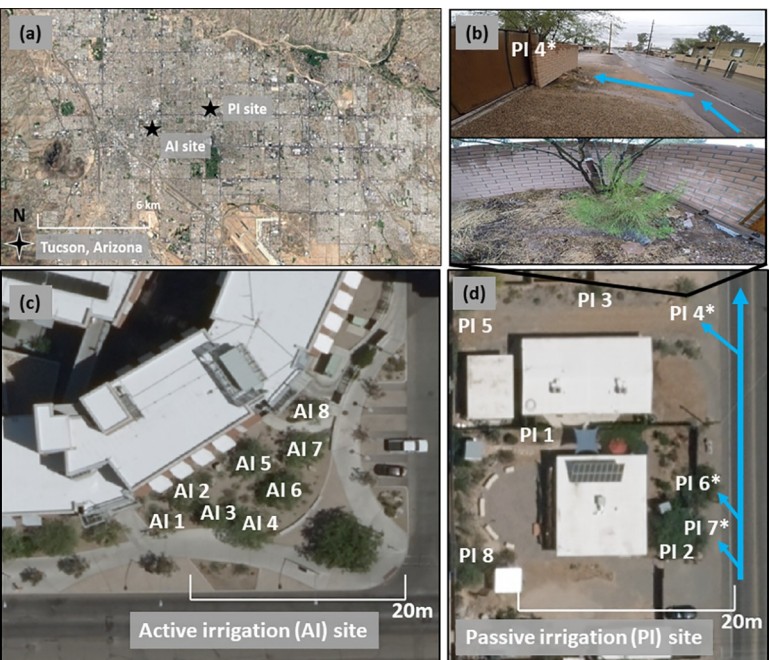

**Fig 2. Study sites.** Aerial photographs were Reprinted from (https://imagery.library.arizona.edu) under a CC BY license, with permission from Pima Association of Governments, original copyright 2015. (a) semiarid city of Tucson, AZ; (b) passive irrigation treatment where a bioswale directs stormwater (blue arrows) into the soil surrounding an urban mesquite tree; (c) the active irrigation site with eight mesquite trees (i.e. AI 1—AI 8); (d) the passive irrigation site with eight mesquite trees (i.e. PI 1—PI 8; asterisks indicate 'PI$_{street}$' trees, which are the trees whose accompanying bioswales are connected to the adjacent road and thus receive appreciable impervious runoff illustrated by blue arrows).

made in the form of PI vs. PI$_{street}$ soil moisture analysis. PI$_{street}$ trees are defined as those trees located within green infrastructure that shared connectivity to the adjacent asphalt street, and hence received appreciable volumes of impervious stormwater runoff (Fig 2B and 2D). PI vs. PI$_{street}$ comparisons are in exclusive reference to their differing soil moisture regimes however, meaning all other references to the PI site (i.e. PI vs. AI cross site comparisons) consider all eight PI trees together.

The AI site is located outside a large residence hall on the University of Arizona campus (32˚13'N, 110˚57'W), roughly two miles southwest of the PI site. The site consists of a garden of eight mesquite trees mixed with various understory shrubs and an exotic evergreen tree (Fig 2C). The residence hall towers to the north with a major street directly to the south, buffered by a sidewalk. Active irrigation occurs once every seven days, replenishing soil moisture losses via a surface dripline system; university managed sensors terminate active irrigation until the following week if sufficient soil moisture is detected. These represent the AI trees of the present study. While active irrigation is the primary water source for the site, several elements of passive water harvesting may provide additional soil moisture to the trees: a) channeled stormwater runoff from the residence hall roof and b) a one-million-gallon collection tank located several hundred meters up the watershed which slowly leaches moisture >1 m below the surface following large rain events. It is undetermined to what degree each of these potential water sources may play in supplementing each tree, but it is assumed that active irrigation supplied at the soil surface is the preeminent source on an annual basis, since it occurs roughly weekly throughout the year. Using the modified pipette method [44], particle size distribution analysis characterized the soil type at both sites as sandy loam.

## Micrometeorology and soil moisture

Each site had a tipping bucket rain gauge (TE-525L, Campbell Scientific Inc.) for precipitation measurement. To record microclimate air temperature, two mesquite trees per site (PI 2, PI 5, AI 6, AI 7) were instrumented with small temperature sensors (ibutton DS1921G Thermochron, Maxim Integrated Products). Sensors were housed in custom-made shields to reflect direct sunlight similar to Crum & Jenerette [45] and suspended in the canopy (2 m in the tree). Three averaged canopy sensors per tree provided air temperature measurements at each site. Soil moisture in response to wetting events was investigated within the root zone profile of each mesquite tree. At both sites, water content reflectometers (CS650, Campbell Scientific Inc.; 5TM, Decagon Devices Inc.) recorded volumetric water content ($VWC$) and soil temperature under eight trees per site (Fig 3A). Sensors were installed horizontally at 10 cm ($VWC_{shallow}$) and 30 cm ($VWC_{deep}$) depths, approximately 1 m on the north side of each tree. Obstructions (i.e. walls, vegetation) that blocked this sensor placement or passive irrigation elements (i.e. basins) that occurred outside of this placement, dictated alternative sensor placement.

## Sap velocity

Granier-style thermal dissipation probes (TDP30, Dynamax Inc.) were used to estimate stem sap velocity of the trees (Fig 3B). Sensor installation and subsequent conversion of raw data to sap flow velocity (Eqs 1 and 2) follow methods from the Dynamax TDP30 user manual and Lu et. al [46]. Two sensors per tree were installed 1 m up the trunk or, in the case of smaller mesquites, only one sensor was installed at a height where the stem diameter was large enough to allow complete insertion of the probe. Sensors were then insulated with reflective wrap and covered with burlap by request of the property owners for aesthetic purposes. Dimensionless flow index ($K$) was found using differential temperature ($\Delta T$) as well as differential

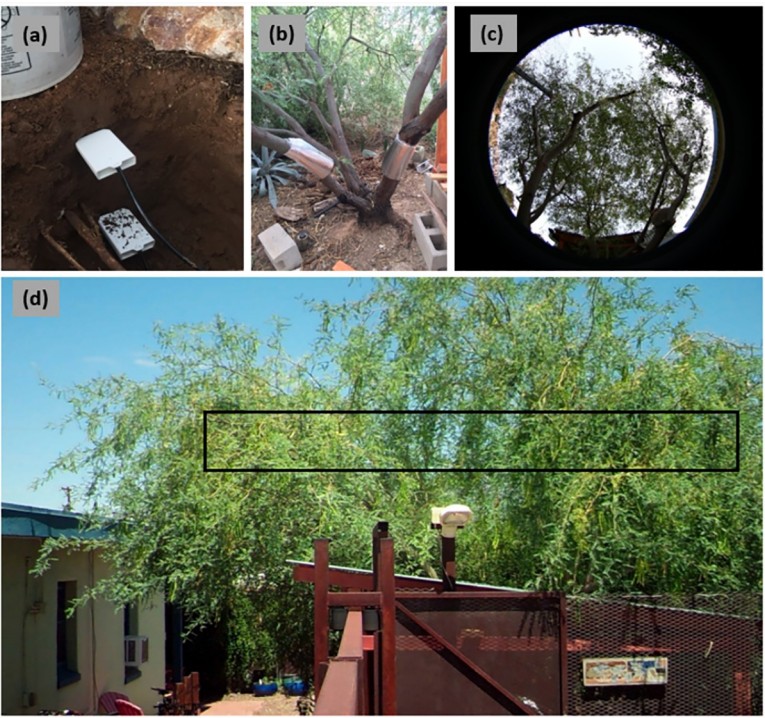

**Fig 3. Data collection methods.** (a) Soil water content reflectometers to measure volumetric water content at 10 cm and 30 cm under each tree; (b) sap flow sensors on mesquite trunks where insulating wrap encloses the thermal dissipation probes; (c) hemispherical photography of mesquite canopy; (d) daily phenocam photo where black box indicates region of interest for greenness analysis.

temperature at zero sap flow ($\Delta TM$, the maximum daily value) (Eq 1).

$$K = (\Delta TM - \Delta T)/\Delta T \qquad (1)$$

Then, average sap velocity ($V_{sap}$) is related to K (Eq 2).

$$V_{sap}(cm/s) = 0.0119 * K^{1.231} \qquad (2)$$

$V_{sap}$ was not converted to sap flow rate because property owners requested that coring or other disturbances to the trees were not administered and so we were unable to determine the conducting sapwood area. We considered site averages of $V_{sap}$ sufficient in understanding the relative trends in cross site water-use dynamics of the trees, since productivity in response to irrigation is the focus here, rather than whole tree transpiration.

## Greenness

Similar to studies from a variety of ecosystems [47,48] we used digital-repeat photography to document canopy phenology dynamics related to the timing and intensity of greenness (Fig 3D). Weatherproof digital cameras (Moultrie M990i Gen2, EBSCO Industries Inc.), or pheno-cams, captured repeat images of individual trees once per day at solar noon. Ideally, pheno-cams were mounted 1 m from the ground and oriented south towards the canopy. Obstructions such as sidewalks required some variable phenocam positioning in order to capture the best view of the canopy. Image analysis followed Richardson et al. [49] by defining a static region of interest for each tree and calculating a greenness index ($I_g$) to quantify the

relative intensities of red-green-blue pixilation of the phenocam JPEG files:

$$I_g = (green - red) + (green - blue)$$
$$= (2 \, {}^*green) - (red + blue) \tag{3}$$

Daily $I_g$ values were then normalized (Eq 4) so that phenocam images could produce a greenness metric ($G_n$) that could be used to compare relative intensities across sites.

$$G_n = (Daily \, I_g - max \, I_g)/(max \, I_g - min \, I_g) \tag{4}$$

## Leaf area index

Leaf area index ($LAI$), or the area of tree canopy per unit ground area, can be estimated using hemispherical photography that captures 180-degree, nadir views of canopy environments [50]. Repeat hemi-photos were taken by establishing a permanent site marker approximately 0.5 m to the south of each tree. Where this camera placement was not possible because of obstructions (e.g. walls; landscaping), permanent markers were established 0.5 m to the north of the tree. At these markers, the camera was mounted to a tripod, oriented towards magnetic north, and levelled so the lens faced directly up. Images were all captured during sunrise, sunset, or uniformly overcast conditions to ensure the best possible contrast between canopy and sky (Fig 3C). Image analysis was done using HemiView software (Delta-T Devices Ltd, Cambridge, UK). HemiView uses geographic location, day of year, orientation of the hemispherical photo, and a binary image gap fraction to calculate $LAI$ and various other canopy and solar indices. Protocol prior to running HemiView included image dating and orientation, and determination of the best representative canopy-sky threshold for individual photos. We obtained a normalized $LAI$ value ($LAI_n$) by setting a minimum and maximum value for site averages.

## Data analysis

Similar to Sanchez-Mejia & Papuga [26] who define separate periods of analysis based on 'wet' and 'dry' seasons, we differentiate a spring and summer growing season because of their predominantly drier and wetter conditions respectively. For the present study, spring was defined as starting April 1 and lasting until the first rain event >10 mm so long as it registered after June 14, at which point it was considered the summer season until October 1 [51]. This was done to address the stark differences in canopy characteristics at both sites that resulted from the onset of the North American Monsoon climatic conditions. Because mesquite trees exhibited cool season deciduousness, the spring and summer periods dominated comparative analysis between sites. Comparisons did not take place over periods where both sites did not register measurements—these data gaps being the result of downed sensors or an inability to collect field measurements (e.g. time series gaps in results section).

Daily averages of shallow and deep soil moisture, sap velocity, and air temperature were calculated from 30-min means and averaged across all sensors for each site. The same was done for daily average $G_n$ and $LAI_n$ with the exception that these were daily and ~monthly photos respectively. A final canopy metric, termed as greenness efficiency ($GE$), we defined as a linear regression between daily averaged $G_n$ and $V_{sap}$. To test for differences between datasets being compared, we used unpaired t tests assuming unequal variance. All analyses including linear regressions were performed in Matlab 2018b (The Mathworks, Inc., Natwick, MA).

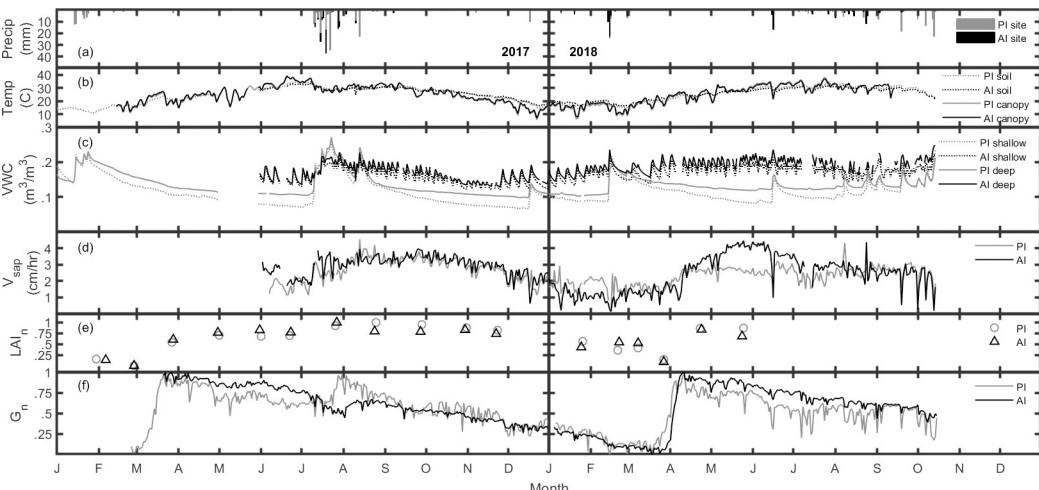

**Fig 4. Daily average time series.** Data from both passive (PI) and active (AI) irrigation sites: (a) precipitation; (b) air and soil temperature; (c) $VWC_{shallow}$ and $VWC_{deep}$; (d) $V_{sap}$; (e) normalized $LAI$; (f) $G_n$.

## Results

### Soil moisture response to irrigation

Soil moisture under passive irrigation (PI) was tied to periods of precipitation while active irrigation (AI) provided weekly artificial inputs that raised moisture conditions regardless of rain events ([Fig 4]). Minimum PI $VWC_{deep}$ and $VWC_{shallow}$ values fell below 0.1 m³/m³ during periods of no rainfall. In contrast, minimum AI $VWC$ values never fell below 0.12 m³/m³ at either depth ([Fig 5]), regardless of long periods without rain. AI provided measurably wetter soil conditions throughout the year than PI (p<0.01), likely as a result of continual irrigation despite dry interstitial periods.

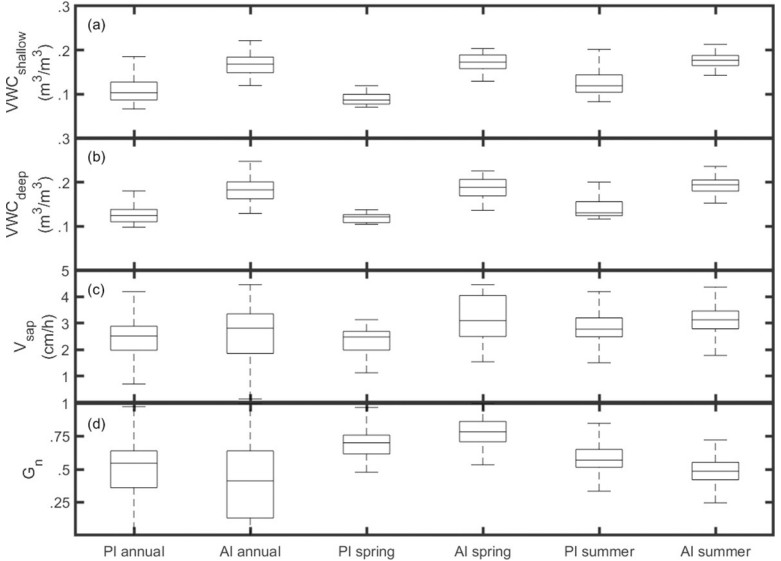

**Fig 5. Seasonal boxplots.** Alternating passive irrigation (PI) and active irrigation (AI) for annual, spring, and summer season (a) $VWC_{shallow}$; (b) $VWC_{deep}$; (c) $V_{sap}$; (d) $G_n$.

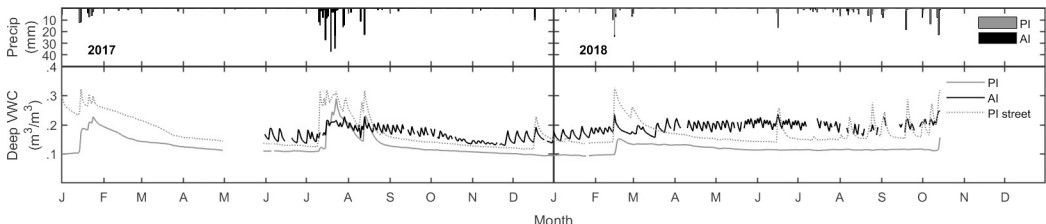

**Fig 6. Deep soil moisture closer look.** Time series of daily average $VWC_{deep}$ values under the PI, PI$_{street}$, and AI treatments.

On average, small rain events (<8 mm) did not raise $VWC_{deep}$ at either site. However, PI soil moisture responses were more reactive depending on the specific type of PI treatment. Fig 6 shows a closer look at $VWC_{deep}$ values with the additional distinction of the passive irrigation 'street' treatments (PI$_{street}$; PI 4, PI 6, PI 7), which received more impervious runoff (see Study Sites section) than the generic PI treatments. The time series illustrates more intense $VWC_{deep}$ responses following rain events in PI$_{street}$ relative to PI treatments which do not receive as much observable impervious runoff. These differences appeared to impact soil moisture conditions over longer timescales as well, whereby PI$_{street}$ plots were significantly wetter (p<0.01) at the deeper soil layer over the entire study period.

## Canopy phenology and greenness

Seasonal climate patterns for the study period were typical for this region of the Sonoran Desert [52]. Following spring warming trends in March 2017 and April 2018, mesquite leaf flush brought an end to the preceding period of canopy dormancy (Fig 4E). Average $LAI$ across sites increased by 0.34 m$^2$/m$^2$ and $G_n$ increased generally from annual low values to annual high values (0 to 1). AI trees increased to maximum $G_n$ just several days after annual lows as a result of new leaves. PI $G_n$ similarly increased following leaf flush, though maximum values did not occur until summer rains. For both years, average spring $G_n$ declined gradually by 0.35 and 0.25 for the PI and AI sites respectively, while $LAI$ remained constant for both sites. In contrast, summer $G_n$ showed distinct cross site differences. In 2017, PI $G_n$ increased from 0.63 to its maximum value during the wet conditions between July 1–31. During this same period, AI $G_n$ decreased from 0.76 to 0.54. Following this drop, AI $G_n$ increased again in early August (up to 0.66) to regain a portion of its earlier $G_n$ value, but a high rainy season peak as observed under PI was never achieved. In 2018, $G_n$ does not diverge in an observable fashion between sites, but rather follows a similar decline following spring peaks. Fig 5D shows PI and AI $G_n$ distributions. While spring $G_n$ is significantly higher for AI trees (p<0.01), summer $G_n$ is significantly higher in PI trees (p<0.01). $LAI_n$ peaked in July and August 2017 for AI and PI respectively; actual values were 1.3 m$^2$/m$^2$ for PI; 1.5 m$^2$/m$^2$ for AI. While AI $LAI$ was greater throughout all seasons, values were not directly compared because of differential tree size, tree stature, and site characteristics. Nonetheless, $LAI$ canopy temporal development was similar across sites.

## Tree sap flow and water use dynamics

Fig 4D shows increased mesquite sap velocity at both sites following spring leaf flush. In 2018, average $V_{sap}$ across sites increased from 1.46 cm/h on March 27 ($LAI$ low) to 2.94 cm/h on April 24 (post leaf flush $LAI$ measurement). As spring progressed, PI $V_{sap}$ remained constant, while AI $V_{sap}$ increased to >4 cm/h in May 2018. Summer $V_{sap}$ varied across years for both sites, whereby the wet summer of 2017 was followed by an increase in $V_{sap}$ relative to the short

period of data collection of spring in the same year. Average $V_{sap}$ for the month of June 2017 was 1.72 cm/h under PI and 2.37 cm/h under AI. With the onset of monsoon rain, average $V_{sap}$ increased to 3.16 cm/h for PI and 3.29 cm/h for AI for the remainder of the summer (July 15-October 1). 2018 by contrast did not experience observable peaks during summer, but rather AI $V_{sap}$ gradually declined to an eventual convergence with the relatively static PI values for the remainder of the year. Fig 5C reflects seasonal differences, whereby $V_{sap}$ for AI was significantly higher compared to PI in both spring and summer seasons ($p<0.01$). Differences over annual timescales still show higher relative $V_{sap}$ for AI but at a lower confidence ($p<0.05$).

Table 1 gives a complete list of Coefficient of Determination ($R^2$) values for linear regressions of both deep and shallow soil moisture relationships of $V_{sap}$ and $G_n$. $R^2$ values are largely similar for $VWC_{deep}$ and $VWC_{shallow}$ for all regressions during the spring and summer seasons; when $R^2$ values are high for $VWC_{deep}$, they are generally high for $VWC_{shallow}$. Differences do occur between the two depths for $V_{sap}$ in the spring, where higher $R^2$ values for $VWC_{deep}$ may suggest stronger soil moisture controls at the deeper layer. For PI, the maximum $VWC_{deep}$ in spring was ~0.14 m³/m³, while summer rains produced a much wider range with a maximum average value of ~0.28 m³/m³ (Fig 7). For AI, $VWC_{deep}$ ranges are similar across spring and summer seasons, both having minimum values ~0.13 m³/m³ and maximum values ~0.24 m³/m³. Despite these differences in deep soil moisture, greenness and sap flow at both sites were more strongly related to $VWC$ in spring than in summer. Fig 7 shows how periods of high $VWC$ and low $V_{sap}$ and $G_n$ in summer likely decreased $R^2$ values for this season at the PI site. Relationships at the AI site are also weaker during summer rather than spring, however this

**Table 1. Regression statistics.** Coefficient of Determination ($R^2$) and slope values resulting from linear regression of $G_n$ and $V_{sap}$ as a function of $VWC_{shallow}$ and $VWC_{deep}$; and $G_n$ as a function of $V_{sap}$ (greenness efficiency; *GE*).

| | Passive Irrigation (PI) | | | Active Irrigation (AI) | | |
|---|---|---|---|---|---|---|
| | $R^2$ | p-value | Slope | $R^2$ | p-value | Slope |
| $G_n$ vs. $VWC_{shallow}$ | | | | | | |
| Annual | 0.001 | <0.01 | 0.18 | 0.10 | <0.01 | 4.87 |
| Spring | 0.60 | <0.01 | 5.13 | 0.20 | <0.01 | 2.06 |
| Summer | 0.27 | <0.01 | 1.96 | 0.01 | <0.01 | 0.63 |
| $G_n$ vs. $VWC_{deep}$ | | | | | | |
| Annual | 0.02 | <0.01 | 1.03 | 0.08 | <0.01 | 3.73 |
| Spring | 0.60 | 0.010 | 5.59 | 0.22 | <0.01 | 1.75 |
| Summer | 0.26 | <0.01 | 2.22 | 0.03 | <0.01 | 1.03 |
| $V_{sap}$ vs. $VWC_{shallow}$ | | | | | | |
| Annual | 0.001 | <0.01 | 0.44 | 0.02 | <0.01 | 6.78 |
| Spring | 0.24 | <0.01 | 16.58 | 0.40 | 0.017 | 26.33 |
| Summer | 0.01 | <0.01 | 1.18 | 0.04 | <0.01 | -6.52 |
| $V_{sap}$ vs. $VWC_{deep}$ | | | | | | |
| Annual | 0.003 | <0.01 | 1.17 | 0.02 | <0.01 | 4.77 |
| Spring | 0.41 | <0.01 | 31.21 | 0.49 | <0.01 | 23.63 |
| Summer | 0.0001 | <0.01 | 0.18 | 0.07 | <0.01 | -8.53 |
| *GE* | | | | | | |
| Annual | 0.30 | <0.01 | 0.17 | 0.50 | <0.01 | 0.18 |
| Spring | 0.27 | <0.01 | 0.01 | 0.12 | <0.01 | 0.03 |
| Summer | 0.15 | <0.01 | 0.10 | 0.05 | <0.01 | 0.03 |

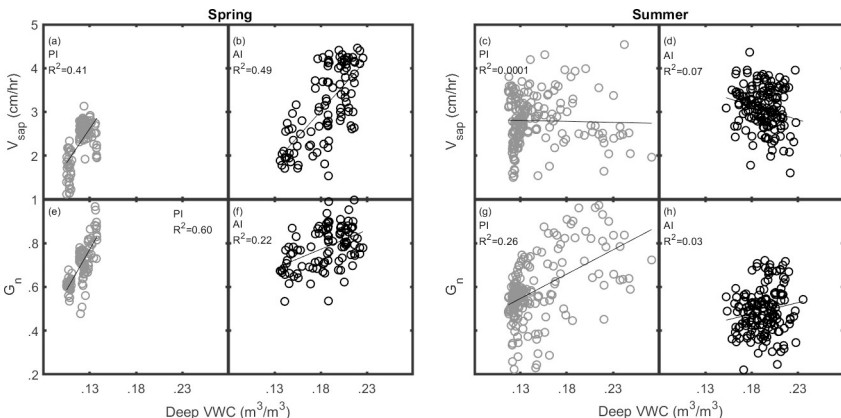

**Fig 7. Deep soil moisture regressions.** Linear regressions between (a-d) $VWC_{deep}$ and $V_{sap}$; (e-h) $VWC_{deep}$ and $G_n$; gray circles are passive irrigation (PI) and black circles are active irrigation (AI). Blocks are separated into spring and summer seasons. All regressions show statistical significance (Table 1).

appears to be driven by a clustering of the data points, rather than opposing high and low values of the dependent variables under short-lived and sporadic high $VWC$ values.

Greenness efficiency ($GE$), or $G_n/V_{sap}$, is plotted separately for PI and AI (Fig 8). Annual data shows a positive relationship, and $GE$ slopes for annual, spring, and summer seasons are shown in Table 1. Similar slopes (PI = 0.17; AI = 0.18) suggest comparable $GE$ for PI and AI on an annual basis, but differences arise between sites in spring and summer. Under PI, spring and summer values frequently overlap at low $V_{sap}$ (around <3 cm/h), while high $V_{sap}$ (>3 cm/h) generally does not occur in spring. Under AI, spring and summer data points generally do not overlap, and a higher range of $V_{sap}$ occurs in spring compared to PI. Spring $GE$ is similar between PI (m = 0.01) and AI (m = 0.03), while summer $GE$ is higher under PI (m = 0.10) than AI (m = 0.03). For AI, similar $GE$ values across seasons are likely a result of a concerted decrease in both $G_n$ and $V_{sap}$ in summer. By comparison, PI had a much sharper $G_n$ increase during the summer of 2017 which may explain the seasonal increase in $GE$.

## Discussion

### Soil moisture under passive and active irrigation

Continuous artificial inputs from active irrigation (AI) maintained higher soil moisture content relative to passive irrigation (PI) throughout the study. PI was subject only to natural precipitation

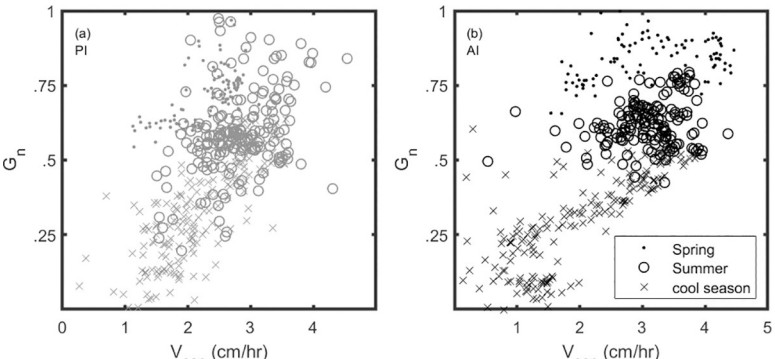

**Fig 8. Greenness efficiency ($GE$).** Linear regression of $G_n$ as a function of $V_{sap}$ for (a) passive irrigation (PI) and (b) active irrigation (AI); dots represent spring; circles represent summer; x's represent the remainder of the year.

events, resulting in significantly lower $VWC_{deep}$ and $VWC_{shallow}$, especially during dry months. Landscape features at the PI site which mimic 'bioswale' and 'rain garden' designs were hypothesized to increase deep moisture content for small rain events (<8 mm) that would otherwise not infiltrate to deep layers. We failed to see this response in most PI treatments. However, passive irrigation defined as $PI_{street}$ (i.e. those bioswales connected to the adjacent paved road) had noticeable increases in deep moisture. Some individual PI treatments in contrast were drier, most likely as a combination of being more disconnected to impervious areas, as well as located in green infrastructure features that were less intense (e.g. smaller basins or swales) than others; PI 2 could be an example of this. Consistent with observations from Houdeshel & Pomeroy [9], $PI_{street}$ treatments which presumably receive more runoff via large contributing impervious areas (the street), had wetter deep soil conditions—even during arid season dry down periods. In support of our hypothesis, several small rain events elicited $VWC_{deep}$ increases at $PI_{street}$ plots when the PI plots located away from the road failed to respond (Fig 6; Dec 2017 and Jan 2018). These results may encourage passive irrigation projects to emphasize stormwater capture from impervious surfaces—such as street curb inlets retrofitted with bioswales.

## Differences in tree phenological activity

Our observations of urban mesquite trees were consistent with natural systems that showed mesquite phenology to be dominated by climate [52,53]. Spring leaf flush and maximum *LAI* coincided with warming temperatures and summer season precipitation respectively (Fig 4E and 4F). Greenness between irrigation types behaved similarly throughout the study and followed a gradual decline after an early spring peak, similar to observations made for other deciduous tree species [47]. Two cross site divergences occurred, however. First and most notably was the greenness peak for PI trees following the extremely wet summer 2017. Second was the elevated and sustained spring greenness levels in AI, during which time PI trees exhibited sharper greenness declines. Though the differential response to summer 2017 rains may obfuscate the interpretation of higher spring greenness in AI, it still suggests that active irrigation (or lack thereof) may influence greenness intensity during this early, dry season.

While greenness indices are imperfect for capturing leaf-level physiology [54], research supports greenness as a tool in understanding carbon uptake dynamics across many plant functional types and ecosystems [47–49,55,56]. Lower spring greenness in PI trees may indicate diminished productivity relative to AI trees during this time. Likewise, high summer greenness in PI following intense summer rains could indicate a period of enhanced carbon uptake. In the case of summer 2017, PI greenness responded to precipitation pulses similar to vegetation in natural desert ecosystems [48]. Conversely, AI greenness mimicked the behavior of trees in water-abundant riparian systems, where precipitation elicits little physiological response to trees with perennial groundwater availability [34,38]. In this vein, it is possible that rain events have diminished influence over urban mesquite tree greenness when active irrigation is already maintaining an elevated soil moisture profile.

## Soil moisture depth and greenness efficiency

On the relative importance of deep versus shallow soil moisture for urban mesquite productivity, linear regressions did not elucidate differences between $VWC_{deep}$ or $VWC_{shallow}$ relationships with $G_n$ or $V_{sap}$. One exception to this may be the spring season $VWC$ vs. $V_{sap}$, where higher $R^2$ values at both sites could indicate slightly stronger relationships at the deep soil layer. General differences between soil depths were small however, and where $VWC_{deep}$ relationships with $V_{sap}$ and $G_n$ were strong, $VWC_{shallow}$ relationships also tended to be strong (Table 1). Because the majority of precipitation events during both spring and summer occurred as successive large

events that wetted both deep and shallow layers, we may attribute this to a lack of small rain events. Under AI, soil moisture responses to irrigation events at both depths closely mimicked each other (Fig 3C and 3D), which likely had a similar effect to a lack of small rain events. In other words, a lack of differential moisture between deep and shallow layers made the partitioning of their differential controls over mesquite productivity less clear than it might have been. Nevertheless, stronger relationships at the 30 cm layer where they occur (e.g. spring season sap flow), suggests that irrigation which emphasizes deep soil moisture availability may be more beneficial for trees in this context.

Spring season $V_{sap}$ and $G_n$ were positively related to soil moisture under both irrigation types, whereas summer relationships were weaker and varied by site (Fig 7). In 2017, $V_{sap}$ under both irrigation types was comparable and remained relatively high throughout the entire post-rain summer season, despite faster soil dry down under PI (Fig 4D; August-October). We expected $V_{sap}$ to be strongly linked to $VWC_{deep}$, but sustained summer sap flow following gradual soil dry down probably weakened this relationship in 2017. Accounting for periods of lagged vegetation response during the wettest periods did not change this. One notable exception of the general conformity between PI and AI $V_{sap}$ however, is the divergence during spring 2018. In this case the assumption is that active irrigation facilitated higher tree water-use during this most arid time of the year.

We coupled $V_{sap}$ and $G_n$ values to define a greenness efficiency (GE) metric, whereby high $G_n$ values occurring simultaneously with low $V_{sap}$ represents a high GE value. $G_n$ generally increased with increasing $V_{sap}$ at both sites (Fig 8), and similar slopes on an annual basis suggest similar GE dynamics over the entire study period. Under PI, GE appeared to increase during the summer season, possibly because of the increased greenness experienced in conjunction with only modest sap flow increases. In contrast, GE did not change for AI between spring and summer seasons, suggesting that high greenness in spring was likely offset by high water-use. This appears to further support differences in the timing of mesquite productivity, possibly as a result of irrigation regime. Thinking in terms of the GE metric adds the implication that changes to the timing of mesquite productivity may also change the efficiency with which trees use available water resources.

Despite the lower AI GE metric in spring, higher greenness at this site in conjunction with regression plots may suggest that increased soil moisture in this season may translate more reliably to increased mesquite productivity. This observation may be connected to similar phenomena in natural ecosystems within the region, where higher mesquite productivity is linked to higher spring soil moisture availability [35], which in turn contributes to higher annual ecosystem carbon uptake [36,37]. Even though productivity in the semiarid Southwest is dominated by summer rains, urban mesquite greenness was less tied to moisture during this season despite using more water; with the important exception of high PI greenness in summer 2017.

## Implications for urban watershed management

Connections exist between our results and future research and management opportunities for urban watersheds. Our hypothesis that soil moisture beyond the reach of rapid evaporation (>20 cm) would be more important than shallow soil moisture for mesquite productivity was not supported. However, the driest soil conditions throughout the study period were less severe at the deep layer under both irrigation types (Fig 4C) which supports our underlying framework that supplying deep moisture may help in avoiding soil water loss via evaporation, which potentially leaves more opportunity for plant uptake.

Our hypothesis that irrigation may alter the timing of mesquite ecohydrology was supported in the cases of specific, isolated periods of time, whereby differential greenness and sap flow

dynamics were observed between the two irrigation types. Actively irrigated mesquite trees maintained higher greenness and sap velocity throughout the dry spring. The same trees however, had a diminished greenness response (despite sustained sap flow) following the summer 2017 rains that created sharp greenness peaks in passively irrigated trees. When this is considered in tandem with regression analysis between soil moisture and productivity metrics, it could be suggested that active irrigation of urban mesquite trees should focus on spring season application. As long as increased water-use is considered an acceptable tradeoff, spring season irrigation may achieve improved greenness, as well as the ecosystem services which are achieved by proxy; e.g. improved aesthetics, growth, and carbon uptake. Likewise, cessation of active irrigation during summer rains until the following year may help avoid diminished water use efficiency, since passively irrigated trees exhibited similar sap velocity but with higher relative greenness during these seasons. While active irrigation ameliorates arid conditions that cause plant stress and reduced productivity, passive irrigation appeared to more closely mimic regional ecohydrological processes and may be sufficient for plants adapted to local climates. As such, green infrastructure that maximizes impervious runoff capture to increase soil moisture could potentially obviate the need for active irrigation and adequately support ecosystem services [9,22].

Balancing the tradeoff between a healthy urban forest and scarce water resources in arid and semiarid cities may be aided through ecohydrological research to inform decisions and maximize net benefits [57]. As a strategy to support urban vegetation while reducing stress on regional water supplies, passive irrigation is of growing interest to many stormwater professionals (e.g. [7]). On the other hand, active irrigation may enhance ecosystem services without relying on unpredictable rain patterns. Future research that could improve upon the work presented here includes: a) coupling soil moisture observations with watershed modelling efforts to better quantify passive irrigation treatments; b) complete sap flow analysis to quantify whole tree water-use under differing irrigation; c) physiological measurements of canopy function such as leaf-level gas exchanges.

## Conclusions

This study compared the ecohydrology of urban mesquite trees in passively and actively irrigated green infrastructure systems in a semiarid city. We observed that green infrastructure such as bioswales that receive stormwater runoff from large impervious surfaces (i.e. city streets) had the best chance of providing additional deep (>20 cm) soil moisture from rain events compared to natural environments. However, we were unable to determine if this deep soil moisture elicited stronger controls over mesquite productivity relative to shallow (<20 cm) moisture. Trees appeared to benefit from active irrigation in spring via increased canopy activity, but once sufficiently rainy conditions arrived in summer, only passively irrigated trees appeared to beneficially use this additional moisture. We hope these observations will provide insight into the management and future research opportunities of urban green infrastructure and ecohydrology in water-limited cities and beyond.

## Acknowledgments

We would like to thank Phil Guertin and Tom Meixner for their guidance throughout this research. Additionally, we thank Yoga Korgaonkar, Neha Gupta, Jack Anderson, Emily Bell, and Liliana Gomez for field and computational assistance. Finally, we thank the staff at Watershed Management Group and University of Arizona Residence Life for providing assistance and the necessary space to conduct this research.

## Author Contributions

**Conceptualization:** Anthony M. Luketich, Shirley A. Papuga.

**Formal analysis:** Anthony M. Luketich.

**Investigation:** Anthony M. Luketich.

**Methodology:** Anthony M. Luketich.

**Supervision:** Shirley A. Papuga.

**Writing – original draft:** Anthony M. Luketich.

**Writing – review & editing:** Anthony M. Luketich, Shirley A. Papuga, Michael A. Crimmins.

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
