## [Decision Letter · Decision Letter 0]

6 Sep 2019

PONE-D-19-21151

Ecohydrology of urban trees under passive and active irrigation in a semiarid city

PLOS ONE

Dear Mr. Luketich,

Thank you for submitting your manuscript to PLOS ONE. After careful consideration, we feel that it has merit but does not fully meet PLOS ONE’s publication criteria as it currently stands. Therefore, we invite you to submit a revised version of the manuscript that addresses the points raised during the review process.

We would appreciate receiving your revised manuscript by Oct 21 2019 11:59PM. To enhance the reproducibility of your results, we recommend that if applicable you deposit your laboratory protocols in protocols.io, where a protocol can be assigned its own identifier (DOI) such that it can be cited independently in the future. For instructions see: http://journals.plos.org/plosone/s/submission-guidelines#loc-laboratory-protocols

We look forward to receiving your revised manuscript.

Kind regards,

Zhihua Wang, Ph.D.

Academic Editor

PLOS ONE

Journal Requirements:

2. We note that Figures in your submission contain map/satellite images which may be copyrighted. All PLOS content is published under the Creative Commons Attribution License (CC BY 4.0), which means that the manuscript, images, and Supporting Information files will be freely available online, and any third party is permitted to access, download, copy, distribute, and use these materials in any way, even commercially, with proper attribution. For these reasons, we cannot publish previously copyrighted maps or satellite images created using proprietary data, such as Google software (Google Maps, Street View, and Earth). For more information, see our copyright guidelines: http://journals.plos.org/plosone/s/licenses-and-copyright.

1.    You may seek permission from the original copyright holder of Figures to publish the content specifically under the CC BY 4.0 license. 

Reviewers' comments:

Reviewer's Responses to Questions

**Comments to the Author**

1. Is the manuscript technically sound, and do the data support the conclusions?

Reviewer #1: Yes

Reviewer #2: Yes

2. Has the statistical analysis been performed appropriately and rigorously? 

Reviewer #1: No

Reviewer #2: No

3. Have the authors made all data underlying the findings in their manuscript fully available?

Reviewer #1: No

Reviewer #2: Yes

4. Is the manuscript presented in an intelligible fashion and written in standard English?

Reviewer #1: Yes

Reviewer #2: Yes

5. Review Comments to the Author

Reviewer #1: The paper is an experimental study on the impact of passive and active irrigation on urban trees. The experiment is well designed and the approach is scientifically sound. Overall the paper is well written and the results are clearly discussed. I only have minor comments:

1) For calculating the Sap velocity, Equation (2) seems to be an empirical relation. Please provide necessary references or clarifications.

2) Figure 6, the R2 values are quite small in some cases, are these linear relationships statistically significant?

3) As this is an experimental study, photos showing the actual in-situ instruments are highly preferred.

4) It might be worthy to include some references on urban irrigation in semi-arid environment, such as Yang and Wang 2015 Optimizing urban irrigation schemes for the trade-off between energy and water consumption, Volo et al. 2015 An ecohydrological approach to conserving urban water through optimized landscape irrigation schedules.

Reviewer #2: Article Review:

Ecohydrology of urban trees under passive and active irrigation in a semiarid city

Recommendation:

Major Revision

General comments:

In the manuscript, the authors used controlled experiment to understand how passive and active irrigation influenced urban trees in the semi-arid city. The overall research and experimental design are complete, and the data analysis and results provide useful insights to the existing literature. However, I would recommend the authors further improve all the figures quality to better support your explanation and revisit the regression analysis to make sure it explains well to the readers. A few typos and grammar errors need to be fixed as well. More detailed comments can be found in the specific comments.

Specific comments:

L48: Can you explain what types of ecosystem services are provided by the green infrastructure? This is too general. Some useful literature is attached here:

Yang, J., & Wang, Z.-H. (2017). Planning for a sustainable desert city: The potential water buffering capacity of urban green infrastructure. Landscape and Urban Planning, 167, 339–347. https://doi.org/10.1016/j.landurbplan.2017.07.014

Zhao, Q., Yang, J., Wang, Z.-H., & Wentz, E. (2018). Assessing the Cooling Benefits of Tree Shade by an Outdoor Urban Physical Scale Model at Tempe, AZ. Urban Science, 2(1), 4. https://doi.org/10.3390/urbansci2010004

Tzoulas, K., Korpela, K., Venn, S., Yli-Pelkonen, V., Kaźmierczak, A., Niemela, J., & James, P. (2007). Promoting ecosystem and human health in urban areas using Green Infrastructure: A literature review. Landscape and Urban Planning, 81(3), 167–178. https://doi.org/10.1016/j.landurbplan.2007.02.001

L53: change “select” to “selected”

L81: similar?? Need more explanation.

L85-87:

Research goals need to be a bit clear. You only say this research applies a framework to the system, which is not very clear. Can you better describe your research objective?

L95: You would also like to explain in one sentence about how your research will contribute to the literature and real-world practise.

L112: Can you make the climate data website as a reference?

L122: What is the highlight for?

L188: Normalized greenness value is more accurate. Normalized value is too general.

L205: Where here is the data analysis section? I don’t understand.

L221: medians of what?

L235-237: What is the p-value here?

L277: change R to R2

L278: the? Or they?

Table 1:

You use acronym quite frequently in your manuscript, such as PI and AI. Please make sure you explain it when it first shows up in the manuscript. Please check all your acronym again.

You might want to change the word “slope” to “coefficient”

Many of your R2 results are very low, and it almost shows no relationship. Also, can you add the p-value for your simple linear regression models?

Although you only have one parameter, you also need to explain what your dependent variable is and what is your independent variable.

I don’t understand your GE part, did you run a linear regression as well here?

Figure:

Two general problems:

1. Figure dpi is too low. You need to increase the figure resolution to at least 300 dpi

2. Figure caption is too long. You can either add them into the main text or make it as a legend in the figure.

Figure 2:

Figure (a) and (b) is quite a crude map. Can you add a city boundary for Tucson? Figure (a) can be a very small inset map next to Figure (b).

You can see an example in the Figure 1 of the listed reference:

Zhao, Q., Yang, J., Wang, Z.-H., & Wentz, E. (2018). Assessing the Cooling Benefits of Tree Shade by an Outdoor Urban Physical Scale Model at Tempe, AZ. Urban Science, 2(1), 4. https://doi.org/10.3390/urbansci2010004

Can you highlight the trail in the figure (c)? That will give reader a better idea.

Figure (d) and (e):

Why don't you use different name for active and passive irrigation site?

Figure 3: I saw there are some gaps in the VWC in 2017 (similar in Figure 5) and LAI in the late 2018. Why does it happen? Does the gap influence the analysis results?

Figure 6: I would recommend to separate summer and spring figures to make the x-axis clearer. And it looks like you have different intervals for your x-axis, I would recommend using the same interval for all the figures.

Figure 7: you will need a legend for this figure, especially when you have different symbols in the same figures.

6. PLOS authors have the option to publish the peer review history of their article (what does this mean?). If published, this will include your full peer review and any attached files.

Reviewer #1: No

Reviewer #2: No

---

## [Author Response · Author response to Decision Letter 0]

21 Oct 2019

We have corrected all major changes mentioned in decision letter including permission forms, format updates, and data repository. In addition, specific reviewer comments are addressed in the 'Response to Reviewer' document which was uploaded to this revision as per the guidelines.

---

## [Editor Report · Decision Letter 1]

23 Oct 2019

Ecohydrology of urban trees under passive and active irrigation in a semiarid city

PONE-D-19-21151R1

Dear Dr. Luketich,

We are pleased to inform you that your manuscript has been judged scientifically suitable for publication and will be formally accepted for publication once it complies with all outstanding technical requirements.

With kind regards,

Zhihua Wang, Ph.D.

Academic Editor

PLOS ONE
---

## [Editor Report · Acceptance letter]

28 Oct 2019

PONE-D-19-21151R1 

Ecohydrology of urban trees under passive and active irrigation in a semiarid city 

Dear Dr. Luketich:

I am pleased to inform you that your manuscript has been deemed suitable for publication in PLOS ONE. Congratulations! Your manuscript is now with our production department. 

With kind regards,

on behalf of

Dr. Zhihua Wang 

Academic Editor

PLOS ONE